SciPost Physics

Submission

# Connecting quasinormal modes and heat kernels in 1-loop determinants

C. Keeler[1], V. L. Martin[1], A. Svesko[1*]

**1** Department of Physics, Arizona State University, Tempe, Arizona 85287, USA *
keelerc@asu.edu, victoria.martin.2@asu.edu, *asvesko@asu.edu

July 18, 2019

## Abstract

We connect two different approaches for calculating functional determinants on quotients of hyperbolic spacetime: the heat kernel method and the quasinormal mode method. For the example of a rotating BTZ background, we show how the image sum in the heat kernel method builds up the logarithms in the quasinormal mode method, while the thermal sum in the quasinormal mode method builds up the integrand of the heat kernel. More formally, we demonstrate how the heat kernel and quasinormal mode methods are linked via the Selberg zeta function. We show that a 1-loop partition function computed using the heat kernel method may be cast as a Selberg zeta function whose zeros encode quasinormal modes. We discuss how our work may be used to predict quasinormal modes on more complicated spacetimes.

# 1 Introduction

Functional determinants of kinetic operators are of interest in theoretical physics because they allow for the study of quantum effects. For example, the 1-loop partition function for a linearized graviton $\phi$ is given by

$$Z^{(1)} = \int D\phi e^{-\frac{1}{2} \int d^D x \sqrt{g} \phi \nabla^2 \phi} = \frac{\det \nabla_v^2}{\sqrt{\det \nabla_s^2 \det \nabla_g^2}}, \tag{1}$$

where $\nabla_s^2$, $\nabla_v^2$ and $\nabla_g^2$ correspond to the kinetic operators for the scalar, vector, and graviton, respectively. Functional determinants also underlie the quantum entropy function [1–3], which encodes quantum corrections to black hole entropies. From a mathematical perspective, functional determinants exhibit the spectral properties of operators on function spaces and provide a classification of smooth manifolds [4]. Here we study two different approaches for computing functional determinants – the heat kernel and quasinormal mode methods – and how they relate to one another via the Selberg zeta function [5].

The first method we explore is the heat kernel method (see e.g. [6]), which directly calculates the spectrum of the kinetic operator of a free quantum field $\psi$ via the eigenvalue equation

$$\nabla^2 \psi_n = \lambda_n \psi_n . \tag{2}$$

This equation can be rewritten in terms of the heat kernel $K$, which solves

$$(\partial_t + \nabla_x^2) K(t; x, y) = 0 , \qquad K(0; x, y) = \delta(x, y) , \tag{3}$$

where $t$ is the heat kernel parameter. The heat kernel is related to the 1-loop correction to the action:

$$S^{(1)} = -\frac{1}{2} \log \det \nabla^2 = -\frac{1}{2} \sum_n \log \lambda_n = \frac{1}{2} \int_{0^+}^{\infty} \frac{dt}{t} \int d^3 x \sqrt{g} K(t; x, x) . \tag{4}$$

For highly symmetric spacetimes, the differential equation (3) can be easily solved. For quotients of such spacetimes, such as thermal AdS and the BTZ black hole, the method of images allows us to calculate the complete heat kernel. This version of the heat kernel method was employed in [7] to calculate the 1-loop determinant of all locally AdS$_3$ spaces.

The second method we consider, put forth in [8, 9], provides additional insight for thermal spacetimes. First, we consider $Z^{(1)}(\Delta)$ as a function of the conformal dimension $\Delta$ in the complex plane. If $Z^{(1)}(\Delta)$ is a meromorphic function, then we can use the Weierstrass factorization theorem to write the 1-loop partition function as a product of zeros and poles, up to an entire function $e^{\text{Pol}(\Delta)}$:

$$Z^{(1)}(\Delta) = e^{\text{Pol}(\Delta)} \frac{\prod_{\Delta_0} (\Delta - \Delta_0)^{d_0}}{\prod_{\Delta_p} (\Delta - \Delta_p)^{d_p}} , \tag{5}$$

where $d_0$ and $d_p$ are the degeneracies of the zeros and poles.

In many cases of interest, $Z^{(1)}(\Delta)$ is indeed a meromorphic function, but not always (cf. flat space[1]). In the case of a scalar field, equation (5) has no zeros. The poles occur at $\Delta_p$ where there is a solution to the Klein Gordon equation

$$\left( -\nabla^2 + \Delta_p (\Delta_p - d) \right) \phi = 0 \tag{6}$$

---

[1]It is fairly straightforward to see why $Z^{(1)}(\Delta)$ is not meromorphic in flat space. Consider the partition function of a scalar field on a two-sphere: $Z^{(1)}(\Delta) = \prod_\ell (\Delta - \frac{\ell(\ell+1)}{a^2})$, where $a$ is the size of the sphere. The flat space limit is obtained by taking the sphere size $a \to \infty$, and we see that the poles accumulate into a branch cut, rendering the partition function non-meromorphic in this limit.

which is smooth and single-valued. In general, these $\Delta_p$ will not correspond to physical mass values. Instead, as [9] shows, the $\Delta_p$ occur when the quasinormal modes $\omega_*(\Delta_p)$ coincide with the Matsubara frequencies. For a static thermal background of temperature $T$, the 1-loop partition function for the scalar field can then be expressed as

$$Z^{(1)}(\Delta) = e^{\mathrm{Pol}(\Delta)} \prod_{n,*} (2\pi i n T - \omega_*(\Delta))^{-1} . \tag{7}$$

The $2\pi i n T$ arises from the Euclidean periodicity condition imposed on the fields, and generalizes to a function $\omega_n(k)$ of the angular momentum quantum number in stationary spacetimes [10].

As mentioned in [7], the subject of computing determinants of Laplacians on Riemannian manifolds is now well-studied among mathematicians [11,12] and physicists [13,14] alike. The so-called Selberg trace formula (and associated Selberg zeta function [14,15]) for a given manifold and kinetic operator calculate the regularized spectrum of that operator. A Selberg zeta function can be assigned to hyperbolic spacetimes of the form $\mathbb{H}^n/\Gamma$, where $\Gamma$ is a discrete subgroup of $SL(2,\mathbb{C})$ [16]. Such quotient spacetimes are the basis for holographic constructions at finite temperature, as well as higher genus generalizations [17].

We show that in the case of $\mathbb{H}^3/\Gamma$, with $\Gamma \simeq \mathbb{Z}$, the Selberg zeta function acts as a bridge between the heat kernel and quasinormal mode methods. This case covers both the rotating BTZ black hole and thermal AdS$_3$ spacetimes. The relationship between the heat kernel and the Selberg zeta function was presented in [13], and [18] (in the context of the static BTZ black hole). The relationship between quasinormal modes and the Patterson-Selberg[2] zeta function was presented in [20,21]. We build on these previous works constructing a connection between the heat kernel and quasinormal methods. Furthering these works, we present a continuous connection between the heat kernel and quasinormal mode methods for calculating 1-loop determinants on hyperbolic quotient spacetimes.

After reviewing the heat kernel and quasinormal mode methods for computing 1-loop partition functions, in Section 2 we explicitly connect the heat kernel and quasinormal mode expressions for both scalar fields and gravitons in the 2+1 dimensional rotating BTZ black hole background. In Section 3 we explain how these methods are formally related through the Selberg zeta function: matching the zeros of the Selberg zeta function to the conformal dimension of the field imposes the condition $\omega_* = \omega_n$. We summarize our conclusions and discuss directions for future work in Section 4.

## 2   Quasinormal Modes and Heat Kernel: Direct Connection

### 2.1   Review of Heat Kernel and Quasinormal Mode Methods

#### 2.1.1   Heat Kernel

We begin by reviewing the heat kernel method for computing functional determinants on a BTZ background, following [7]. For a real massive scalar field of mass $m$, the heat kernel

---

[2]There is a technical difference between Selberg and Patterson-Selberg zeta functions. In hyperbolic quotient spacetimes in which the volume of the fundamental domain is finite $\mathrm{Vol}(\mathcal{F}) < \infty$, one can assign a Selberg zeta function to that spacetime in a more or less straightforward way (cf [14]). The $\mathrm{Vol}(\mathcal{F}) = \infty$ case is more complicated, but in spacetimes that retain a high degree of symmetry (such as the BTZ black hole) it is still possible to assign a zeta function to this quotiented spacetime, and this is referred to as the Patterson-Selberg zeta function [16,19]. From here on, when we say the Selberg zeta function, it is implied that we mean the Patterson-Selberg zeta function.

over $\mathbb{H}^3$ can be found by directly solving the heat kernel differential equation (3):

$$K^{\mathbb{H}^3}(t;r) = \frac{1}{(4\pi t)^{3/2}} \frac{r}{\sinh r} \exp\left(-(m^2+1)t - \frac{r^2}{4t}\right) . \tag{8}$$

The 1-loop contribution to the effective action then becomes

$$\begin{aligned}
S^{(1)} &= -\frac{1}{2}\log\det(-\nabla^2 + m^2) = \frac{1}{2}\int_0^\infty \frac{dt}{t} \int d^3x \sqrt{g} K^{\mathbb{H}^3}(t;x,x) \\
&= \frac{1}{2}\mathrm{Vol}(\mathbb{H}^3) \int \frac{dt}{t} \frac{e^{-(m^2+1)t}}{(4\pi t)^{3/2}} \\
&= \frac{\mathrm{Vol}(\mathbb{H}^3)}{12\pi}(m^2+1)^{3/2} ,
\end{aligned} \tag{9}$$

where the $\mathrm{Vol}(\mathbb{H}^3)$ represents an IR divergence coming from the integral over $\mathbb{H}^3$. As noted in [7], the UV divergence (corresponding to $t \to 0$) can be removed by performing an analytic continuation of the $t$ integral.

The heat kernel is well suited to compute functional determinants on quotient spaces of the form $\mathcal{M} = \mathbb{H}^3/\Gamma$, including the BTZ black hole and thermal AdS where $\Gamma \simeq \mathbb{Z}$. Since the heat kernel differential equation (3) is linear, the heat kernel on $\mathbb{H}^3/\Gamma$ can be determined using the method of images:

$$K^{\mathbb{H}^3/\Gamma}(t;x,y) = \sum_{\gamma \in \Gamma} K^{\mathbb{H}^3}(t;x,\gamma y) . \tag{10}$$

For the BTZ and thermal AdS cases, the heat kernel on $\mathbb{H}^3/\mathbb{Z}$ becomes:

$$K^{\mathbb{H}^3/\mathbb{Z}}(t;x,y) = \sum_{k \in \mathbb{Z}} K^{\mathbb{H}^3}(t;r(x,\gamma^k x')) . \tag{11}$$

A detailed account of how to find $\gamma$ from the group theoretic structure of the BTZ black hole is given in Appendix A.

From (11) the scalar 1-loop determinant on $\mathbb{H}^3/\mathbb{Z}$ is

$$\begin{aligned}
-\log\det\mathcal{O} &= \int_0^\infty \frac{dt}{t} \int d^3x \sqrt{g} K^{\mathbb{H}^3/\mathbb{Z}}(t;x,x) \\
&= \mathrm{Vol}(\mathbb{H}^3/\mathbb{Z}) \int_0^\infty \frac{dt}{t} \frac{e^{-(m^2+1)t}}{(4\pi t)^{3/2}} + \sum_{k\neq 0} \int_0^\infty \frac{dt}{t} \int_{\mathbb{H}^3/\mathbb{Z}} d^3x \sqrt{g} K^{\mathbb{H}^3}(t;r(x,\gamma^k x)) ,
\end{aligned} \tag{12}$$

with $\mathcal{O} = (-\nabla^2 + m^2)$. The explicit functional determinant found in [7] is

$$-\log\det\mathcal{O} = \sum_{k=1}^\infty \frac{e^{-2\pi k \tau_2 \sqrt{1+m^2}}}{2k|\sin\pi k\tau|^2} = 2\sum_{k=1}^\infty \frac{|q|^{k\Delta}}{k|1-q^k|^2} , \tag{13}$$

where $q \equiv e^{2\pi i\tau}$, $\tau = \tau_1 + i\tau_2$, and $\Delta = (1 + \sqrt{1+m^2})$. In (13) we have dropped the $k = 0$ contribution, as this infinite volume term can be removed via a counterterm.

Evaluating the sum over images $k$, the 1-loop partition function for a real scalar field in a rotating BTZ black hole may be expressed as a product over integers $\tilde{\ell}, \tilde{\ell}'$

$$Z_{\text{scalar}}^{(1)}(\tau,\bar\tau) = (\det\mathcal{O})^{-1/2} = \prod_{\tilde{\ell},\tilde{\ell}'=0}^\infty \frac{1}{1 - q^{\tilde{\ell}+\Delta/2}\bar{q}^{\tilde{\ell}'+\Delta/2}} . \tag{14}$$

In a similar manner, the graviton 1-loop partition function on $\mathbb{H}^3/\mathbb{Z}$ is [7]:

$$Z_{\text{grav}}^{(1)}(\tau,\bar\tau) = \prod_{m=2}^\infty \frac{1}{|1 - q^m|^2} . \tag{15}$$

### 2.1.2 Quasinormal Modes

We now briefly outline how to derive the 1-loop partition function via the quasinormal mode method, following [10]. We restrict ourselves to a real scalar field $\varphi$ in a rotating BTZ black hole background. The quasinormal modes for higher spin $s$ fields in this background were computed in [22]. The scalar field behavior near the Euclidean horizon $r \sim r_+$ is

$$\varphi(\xi, T_E, \Phi) \sim \xi^{\pm ik_T} e^{-k_T T_E} e^{-i\ell\Phi} \ . \tag{16}$$

Here $k_T = \frac{\omega r_+ - i\ell|r_-|}{r_+^2 + |r_-|^2}$ is the frequency conjugate to the Euclidean time coordinate $T_E$, $\ell \in \mathbb{Z}$ is the angular momentum quantum number conjugate to the coordinate $\Phi$ and $\xi$ is a radial coordinate defined in Appendix A.

Periodicity of $\varphi$ in the $T_E$ direction requires that $k_T = in$ for $n \in \mathbb{Z}$. Therefore, solutions (16) will only occur at specific quantized values $\omega_n$[3]:

$$-ik_T = n \Rightarrow \frac{\omega_n}{2\pi} = 2i\frac{T_L T_R}{T_L + T_R}n + \frac{T_R - T_L}{T_L + T_R}\frac{\ell}{2\pi} \ , \tag{17}$$

with $T_{L,R} = \frac{1}{2\pi}(r_+ \mp r_-)$. From (16), we can see that these values correspond to the ingoing (quasinormal, $n > 0$) and outgoing (antiquasinormal, $n < 0$) modes, once we Wick-rotate to real time. This equation can be rewritten as

$$\frac{\omega_n}{2\pi} = \frac{in}{2}(T_R + T_L) - \frac{(T_R - T_L)}{2}k_\Phi(n, \ell) \ , \quad k_\Phi(n, \ell) = \frac{(T_R - T_L)}{(T_R + T_L)}in - \frac{\ell}{\pi(T_R + T_L)} \ , \tag{18}$$

where we have introduced $k_\Phi(n, \ell)$ following the notation of [10].

The ingoing quasinormal mode frequencies of a real scalar field on a rotating BTZ black hole background are given by [23]

$$\omega_* = -\ell - 2\pi iT_R(2p + \Delta) \ , \quad \omega_* = \ell - 2\pi iT_L(2p + \Delta) \ , \tag{19}$$

while the outgoing antiquasinormal frequencies are given by

$$\omega_* = -\ell + 2\pi iT_R(2p + \Delta) \ , \quad \omega_* = \ell + 2\pi iT_L(2p + \Delta) \ . \tag{20}$$

Here $p \in \mathbb{N}$ and $\ell \in \mathbb{Z}$.

If $Z^{(1)}(\Delta)$ is a meromorphic function, then it may be written as a product over its poles and zeros – a consequence of the Weierstrass factorization theorem (5). For a real scalar field, $Z^{(1)} \propto (\det\nabla_s^2)^{-1/2}$, and thus has no zeros. The determinant will have a zero, and thus $Z^{(1)}$ will have a pole, whenever the operator $\nabla_s^2$ has a zero mode. These zero modes occur when $\Delta$ is tuned such that the Klein-Gordon equation (6) has a smooth, single-valued solution for $\varphi$ in Euclidean signature which obeys the asymptotic boundary conditions. We will denote these "good $\varphi$" by $\varphi_{*,n}$, where $n$ labels the mode number in the Euclidean time direction and $*$ represents all other quantum numbers characterizing the solution. The associated $\Delta$ for which $\varphi_{*,n}$ solve the Klein-Gordon equation are likewise denoted as $\Delta_{*,n}$. Therefore, poles in $Z^{(1)}(\Delta)$ occur whenever $\Delta = \Delta_{*,n}$.

The central result of [9] is the relationship between these Euclidean zero modes $\varphi_{*,n}$ and the Lorentzian quasinormal modes, achieved via a Wick rotation. Quasinormal modes are Lorentzian modes which have purely ingoing behavior at the horizon and satisfy the asymptotic boundary conditions. Both conditions are only satisfied at a discrete set of frequencies $\omega_*(\Delta)$. For physical $\Delta$, these frequencies generically have both real and imaginary

---

[3]We can interpret these $\omega_n$ as "Matsubara frequencies" in a thermal field theory with rotation, where the integer $\ell$ corresponds to the rotation.

parts, so quasinormal modes are the damped modes indicating the ringdown of the black hole. Antiquasinormal modes instead satisfy purely outgoing behavior at the horizon.

When $\Delta$ is tuned to $\Delta_{*,n}$, we find

$$\omega_*(\Delta_{*,n}) = \omega_n \ . \tag{21}$$

That is, setting $\Delta = \Delta_{*,n}$ aligns the quasinormal modes with the Matsubara frequencies. For static spacetimes the Matsubara frequencies are $\omega_n = 2\pi i n T$; for stationary spacetimes they are given by (17). Consequently, if we know all of the (anti)quasinormal frequencies as a function of $\Delta$, then we can immediately locate the poles in $Z^{(1)}(\Delta)$: they are where $\Delta$ is tuned such that $\omega_*(\Delta) = \omega_n$.

Using this insight, the 1-loop determinant of a real scalar field on a rotating BTZ black hole background is given by [10]

$$
\begin{aligned}
\left(\frac{e^{\text{Pol}}}{Z^{(1)}}\right)^2 = &\prod_{n>0,p\geq 0,\ell} (\omega_n + \ell + 2\pi i T_R(2p + \Delta))(\omega_n - \ell + 2\pi i T_L(2p + \Delta)) \\
&\prod_{n<0,p\geq 0,\ell} (\omega_n + \ell - 2\pi i T_R(2p + \Delta))(\omega_n - \ell - 2\pi i T_L(2p + \Delta)) \\
&\prod_{p\geq 0,\ell} (\omega_0 + \ell + 2\pi i T_R(2p + \Delta))(\omega_0 - \ell + 2\pi i T_L(2p + \Delta)) \ .
\end{aligned}
\tag{22}
$$

After some algebraic manipulation and absorbing a factor into $\text{Pol}(\Delta)$ we obtain

$$Z^{(1)} = e^{\text{Pol}(\Delta)} \prod_{\tilde{\ell},\tilde{\ell}'=0}^{\infty} \frac{1}{(1 - q^{\tilde{\ell}+\Delta/2}\bar{q}^{\tilde{\ell}'+\Delta/2})} \ , \tag{23}$$

where $q \equiv e^{-2\pi(2\pi T_L)} = e^{2\pi i\tau}$, $\bar{q} \equiv e^{-2\pi(2\pi T_R)} = e^{-2\pi i\bar{\tau}}$, $\tau = 2\pi i T_L$ and $\bar{\tau} = -2\pi i T_R$. In a similar fashion, [10] wrote the 1-loop partition function for the graviton in the same background as

$$Z_{\text{grav}}^{(1)} = \prod_{\tilde{\ell}=0}^{\infty} \frac{1}{(1 - q^{\tilde{\ell}+2})(1 - \bar{q}^{\tilde{\ell}+2})} \ . \tag{24}$$

Before we move on to show the explicit *connection* between the heat kernel and quasinormal mode methods of computing 1-loop determinants, we highlight how they are *different*. While the heat kernel method makes explicit use of the group theoretic structure of the quotient spacetime via the method of images, the quasinormal mode method does not. More specifically, with the heat kernel method we used that $\mathbb{H}^3/\Gamma$ is locally $AdS_3$ and that the identifications $(t_E, \phi) \sim (t_E, \phi+2\pi)$ and $(t_E, \phi) \sim (t_E+\beta_0, \phi+\theta)$ easily determine the group element $\gamma \in \Gamma$. The quasinormal mode method, on the other hand, requires a detailed knowledge of the quasinormal modes for a field $\varphi$ on the thermal background. Nonetheless, the two approaches agree exactly, cf. (22), (14), and (24), (15). This agreement suggests that the quasinormal modes contain information about the group theoretic structure of the orbifold geometry $\mathbb{H}^3/\Gamma$, and that the heat kernel encodes the quasinormal modes.

The two methods also differ in how the modular transformation $\tau_{\text{BTZ}} \rightarrow -1/\tau_{\text{TAdS}}$ manifests when comparing the thermal $AdS_3$ and BTZ calculations. In the heat kernel method, $Z^{(1)}$ for thermal AdS is obtained by performing the modular transformation at almost any point in the calculation. In the quasinormal mode method, the relationship between the BTZ and thermal AdS 1-loop partition functions via modular transformation is not apparent until the end of the calculation, as individual normal modes of thermal

AdS do not directly map to individual quasinormal modes of BTZ. The fact that the collection of normal modes or quasinormal modes lead to a 1-loop partition function of the same form, which also matches the results determined using the heat kernel method, again suggests that the modes contain critical information about the underlying quotient structure of the spacetime.

## 2.2 An Explicit Connection: Scalars

We now explicitly compare the calculations of the 1-loop determinant of a real scalar field in a rotating BTZ black hole background via the quasinormal mode and heat kernel methods. We will find (1) the image sum (11) builds up the logarithm appearing in the quasinormal mode expressions (25), and (2) the thermal mode sum builds up the integrand appearing in the heat kernel expression (12).

For calculational ease we study the logarithm of the determinant. From the quasinormal mode method we have [10]

$$
\log Z^{(1)} - \text{Pol}(\Delta) = - \sum_{n>0, p\geq 0} \left\{ \log(1 - q^{n+p}\bar{q}^p (q\bar{q})^{\Delta/2}) + \log\left(1 - \bar{q}^{n+p}q^p (q\bar{q})^{\Delta/2}\right) \right\}
$$
$$
- \sum_{p\geq 0} \log\left(1 - (q\bar{q})^{p+\Delta/2}\right) ,
\tag{25}
$$

where $p \in \mathbb{N}$ labels a particular quasinormal mode and $n \in \mathbb{Z}$ labels the thermal integer in the regularity condition $k_T = in$. We expand the logarithm by introducing a new integer $k$ via

$$
\log(1 - x) = - \sum_{k=1}^{\infty} \frac{x^k}{k} .
\tag{26}
$$

As we will see shortly, $k$ has a physical interpretation: it is the index in the image sum used in the heat kernel method.

Implementing (26), (25) becomes

$$
\log Z^{(1)} - \text{Pol}(\Delta) = \sum_{k=1}^{\infty} \frac{(q\bar{q})^{k\Delta/2}}{k} \left[ \sum_{n=1}^{\infty} \left(q^{kn} + \bar{q}^{kn}\right) + 1 \right] \sum_{p\geq 0} (q\bar{q})^{kp}
$$
$$
= \sum_{k=1}^{\infty} \frac{(q\bar{q})^{k\Delta/2}}{k} \left[ \frac{1 - (q\bar{q})^k}{(1 - q^k)(1 - \bar{q}^k)} \right] \frac{1}{(1 - (q\bar{q})^k)}
\tag{27}
$$
$$
= \sum_{k=1}^{\infty} \frac{(q\bar{q})^{k\Delta/2}}{k} \left[ \sum_{\tilde{\ell}, \tilde{\ell}'=0}^{\infty} q^{k\tilde{\ell}} \bar{q}^{k\tilde{\ell}'} \right] ,
$$

where we used a geometric series to get the final line. This expression exactly matches the result in [7] for a scalar field in thermal AdS, after sending $\tau_{\text{BTZ}} \to -1/\tau_{\text{AdS}}$ in $q$.

From the first line of (27), we can split our expression into three factors: one involving the thermal number $n$, one involving the mode number $p$, and another factor that only depends on the index $k$ coming from the Taylor expansion of the logarithm. Comparing to functional determinant as found from the heat kernel (13), we see that $k$ is exactly the image index in the method of images.

In the last line of (27) there are two terms. The term in brackets is obtained by carrying out the sums over quasinormal modes, while the other term is independent of $n$ and $p$. Both terms have distinct physical interpretations in the heat kernel picture. Multiplying

and dividing (27) by $(q\bar{q})^{-k/2}$ we obtain

$$\log Z^{(1)} - \mathrm{Pol}(\Delta) = \sum_{k=1}^{\infty} \frac{(q\bar{q})^{k/2(\Delta-1)}}{k} \left[ \frac{1}{(q^{-k/2} - q^{k/2})(\bar{q}^{-k/2} - \bar{q}^{k/2})} \right]. \tag{28}$$

Evaluating the term in brackets we find

$$((q^{-k/2} - q^{k/2})(\bar{q}^{-k/2} - \bar{q}^{k/2}))^{-1} = (4|\sin(\pi k \tau)|^2)^{-1}, \tag{29}$$

and using that

$$(q\bar{q}) = e^{2\pi i(\tau-\bar{\tau})} = e^{-4\pi\tau_2} , \tag{30}$$

and $\Delta - 1 = \sqrt{1 + m^2}$, we arrive at

$$\log Z^{(1)} - \mathrm{Pol}(\Delta) = \sum_{k=1}^{\infty} \frac{e^{-2\pi\tau_2 k\sqrt{m^2+1}}}{k} \left[ \frac{1}{4|\sin(\pi k \tau)|^2} \right] . \tag{31}$$

This is precisely the method of images result for the 1-loop partition function, $(14)$[4]. In summary, the sum over images in the heat kernel method builds the series expansion for the logarithms in the quasinormal mode method. Conversely, the combined sums over the thermal number $n$ and radial number $p$ in the quasinormal mode method build the integration measure and the heat kernel in (12).

## 2.3 An Explicit Connection: Gravitons

Similarly, the quasinormal mode method of computing the 1-loop partition function for massless 3-dimensional gravitons rebuilds the associated heat kernel on a $\mathbb{H}^3/\mathbb{Z}$ background. The form of the graviton 1-loop determinant is [24]

$$Z_{\mathrm{grav}}^{(1)} = \left( \frac{\det_T(-\nabla^2 + 2/L^2)}{\det_{STT}(-\nabla^2 - 2/L^2)} \right)^{1/2} , \tag{32}$$

where the denominator is the determinant for symmetric, transverse and traceless rank-2 tensors and the numerator is the determinant for transverse vector fields. The numerator is a ghost contribution due to the presence of extra gauge redundancies associated with massless 3-dimensional gravitons. If we use the same steps which led us to (31), we recover the heat kernel for a massless 3-dimensional graviton living on a (rotating) BTZ background, as we now show explicitly.

Following the quasinormal mode method presented in [10][5], the logarithm of the 1-loop partition function for the massless graviton becomes

$$\log Z_{\mathrm{grav}}^{(1)} = \sum_{\tilde{\ell},\tilde{\ell}'=0}^{\infty} \log\left[ \left(1 - q^{\tilde{\ell}+2}\bar{q}^{\tilde{\ell}'+1}\right)\left(1 - q^{\tilde{\ell}+1}\bar{q}^{\tilde{\ell}'+2}\right) \right] - \sum_{\tilde{\ell},\tilde{\ell}'=0}^{\infty} \log\left[ \left(1 - q^{\tilde{\ell}+2}\bar{q}^{\tilde{\ell}'}\right)\left(1 - q^{\tilde{\ell}}\bar{q}^{\tilde{\ell}'+2}\right) \right] . \tag{33}$$

---

[4]We can also compare (31) to (13). The factor of two difference arises because (13) is the functional determinant of the kinetic operator, while (31) is the logarithm of the 1-loop partition function, and $Z^{(1)} = (\det \mathcal{O})^{-1/2}$. Had we considered a complex scalar field instead of a real scalar field, there would not be an additional factor of $1/2$ in $Z^{(1)}$.

[5]We impose the same condition as for the scalar fields in (3.2), except that the conformal dimension $\Delta \to \Delta_2 \pm 2\frac{T_R - T_L}{T_R + T_L}$ where $\Delta_2$ is the conformal dimension of the graviton. A similar argument holds for the calculation of the quasinormal mode spectrum for massive spin-1 vector fields, except here the replacement is $\Delta \to \Delta_1 \pm \frac{T_R - T_L}{T_R + T_L}$. These shifts occur because some low-mode number quasinormal modes do not Wick-rotate to well-behaved Euclidean modes, as explained in the appendix of [10].

The first sum on the right hand side is

$$-\sum_{\tilde{\ell},\tilde{\ell}'=0}^{\infty} \log\left[\left(1-q^{\tilde{\ell}+2}\bar{q}^{\tilde{\ell}'+1}\right)\left(1-q^{\tilde{\ell}+1}\bar{q}^{\tilde{\ell}'+2}\right)\right] = \sum_{k=1}^{\infty}\sum_{\tilde{\ell},\tilde{\ell}'=0}^{\infty} \frac{1}{k}(q^{k(\tilde{\ell}+2)}\bar{q}^{k(\tilde{\ell}'+1)} + q^{k(\tilde{\ell}+1)}\bar{q}^{k(\tilde{\ell}'+2)})$$

$$= \sum_{k=1}^{\infty} \frac{1}{k|1-q^k|^2}(q^{2k}\bar{q}^k + \bar{q}^{2k}q^k)$$

$$= \sum_{k=1}^{\infty} \frac{|q|^{2k}(q^k+\bar{q}^k)}{k|1-\bar{q}^k|^2} \;,$$

$$(34)$$

while the second sum on the right hand side of (33) is

$$-\sum_{\tilde{\ell},\tilde{\ell}'=0}^{\infty} \log\left[\left(1-q^{\tilde{\ell}+2}\bar{q}^{\tilde{\ell}'}\right)\left(1-q^{\tilde{\ell}}\bar{q}^{\tilde{\ell}'+2}\right)\right] = \sum_{k=1}^{\infty}\sum_{\tilde{\ell},\tilde{\ell}'=0}^{\infty} \frac{1}{k}(q^{k(\tilde{\ell}+2)}\bar{q}^{k\tilde{\ell}'} + q^{k\tilde{\ell}}\bar{q}^{k(\tilde{\ell}'+2)})$$

$$= \sum_{k=1}^{\infty} \frac{q^{2k}+\bar{q}^{2k}}{k|1-q^k|^2} \;.$$

$$(35)$$

Combining (34) and (35) leads to

$$\log Z_{\text{grav}}^{(1)} = \sum_{k=1}^{\infty} \frac{q^{2k}+\bar{q}^{2k}-|q|^{2k}(q^k+\bar{q}^k)}{k|1-\bar{q}^k|^2}$$

$$= \int_0^{\infty} \frac{dt}{t}\sum_{k=1}^{\infty} \frac{2\pi^2\tau_2}{|\sin(\pi k\tau)|^2}\frac{e^{-(2\pi k\tau_2)^2/4t}}{4\pi^{3/2}\sqrt{t}}\left[e^{-t}\cos(4\pi k\tau_1) - e^{-4t}\cos(2\pi k\tau_1)\right] \;,$$

$$(36)$$

matching the full 1-loop free energy given in section 4, equation (4.27), in [7]. We can easily rewrite this quantity as

$$\log Z_{\text{grav}}^{(1)} = \frac{1}{2}\int_0^{\infty} \frac{dt}{t}(K^{(2)}(\tau,\bar{\tau};t)e^{2t} - K^{(1)}(\tau,\bar{\tau};t)e^{-2t}) \;,$$

$$(37)$$

where we have introduced the heat kernel $K^{(s)}(\tau,\bar{\tau};t)$ for fields of spin-$s$ [25]:

$$K^{(s)}(\tau,\bar{\tau};t) = \sum_{k=1}^{\infty} \frac{2\pi\tau_2}{\sqrt{4\pi t}|\sin k\pi\tau|^2}\cos(2\pi sk\tau_1)e^{\frac{-(2k\pi\tau_2)^2}{4t}}e^{-(s+1)t} \;.$$

$$(38)$$

The sum over images in the heat kernel (38) method rebuilds the power series expansion for $\log(1-x)$ in the quasinormal mode method, and the combined sum over the radial quantum number $p$ and thermal integer $n$ in the quasinormal mode method build the integrand of the heat kernel in (37).

# 3  Quasinormal Modes and Heat Kernel: Formal Connection

Inspired by the results in the previous section, we now seek a more formal connection between the quasinormal mode and heat kernel methods for calculating functional determinants of Laplacians. The connection resides in the Selberg zeta function $Z_\Gamma$, a quantity that can be assigned to a quotient spacetime $\mathcal{M}/\Gamma$ with $\Gamma$ a discrete subgroup of $SL(2,\mathbb{C})$.

For concreteness, we again work in the case of a real massive scalar field in the rotating BTZ black hole background[6]. The graviton result is worked out subsequently.

The quantity $\det(\nabla^2)$ is divergent as $\nabla^2$ is an unbounded operator, so our first task in computing its spectrum is to regularize it. Though various regularization schemes are possible (see e.g. [14] and references therein), we will see that zeta function regularization yields a particularly simple expression for the finite piece of $Z^{(1)}$ in terms of the generalized Riemann zeta function [26].

We can assign a generalized Riemann zeta function $\zeta(s)$ to an invertible operator $\mathcal{O}$ satisfying $\mathcal{O}\psi_n(x) = \lambda_n \psi_n(x)$, defined by

$$\zeta(s|\mathcal{O}) \equiv \sum_n \frac{1}{\lambda_n^s} \ . \tag{39}$$

This function is similar to the standard Riemann zeta function, except the sum is taken over the nonzero eigenvalues of $\mathcal{O}$ rather than over the natural numbers. This zeta function allows us to neatly express the functional determinant $\det \mathcal{O}$ as

$$\log \det \mathcal{O} = - \frac{\partial \zeta(s|\mathcal{O})}{\partial s}\bigg|_{s=0} \ , \tag{40}$$

and thus the generalized Riemann zeta function is a repackaging of $\det \mathcal{O}$ itself.

If we also introduce the heat kernel associated with $\mathcal{O}$,

$$K(t; x, x'|\mathcal{O}) = \sum_n e^{-\lambda_n t}\psi_n(x)\psi_n^*(x') \ , \tag{41}$$

we see that the generalized zeta function is given by the Mellin transform of this heat kernel [14]:

$$\zeta(s|\mathcal{O}) = \frac{1}{\Gamma(s)} \int_0^\infty t^{s-1}\mathrm{Tr}K(t; x, x|\mathcal{O})dt \ . \tag{42}$$

Here $\Gamma(s)$ is the gamma function and the trace implies an integral over $K_t(x, x|\mathcal{O})$ with respect to $x$.

## 3.1 A Formal Connection: Scalars

We now specialize to the case of a real massive scalar field in a rotating BTZ black hole background in order to explore the link between the generalized Riemann zeta function and the Selberg zeta function. This specialization will facilitate comparison with the results of the previous section[7] . Recall that the 1-loop determinant for such a scalar field is, as in (12)-(13):

$$\begin{aligned}
-\log \det \nabla_s^2 &= \zeta'^{(e)} + \sum_{k \neq 0} \int_0^\infty \frac{dt}{t} \int_{\mathbb{H}^3/\mathbb{Z}} d^3x \sqrt{g} K^{\mathbb{H}^3}(t; r(x, \gamma^k x)) \\
&= \zeta'^{(e)} + \sum_{k=1}^\infty \frac{e^{-2\pi k \tau_2 \sqrt{1+m^2}}}{2n|\sin \pi k\tau|^2} \ .
\end{aligned} \tag{43}$$

The term $\zeta'^{(e)}$ represents the $k = 0$ term of the sum. It is a divergent quantity proportional to the volume of $\mathbb{H}^3/\mathbb{Z}$, and its superscript is to remind us that it corresponds to the identity element of the group action $\Gamma$.

---

[6]The calculation that we present is equivalent to that of thermal AdS, with the appropriate redefinition of $q$, arising from the modular transformation $\tau_{\mathrm{BTZ}} \to -1/\tau_{AdS}$.

[7]To generalize this comparison to higher spin fields, see subsection 3.2 for the graviton case and [27] for spinor fields.

As reviewed in Appendix A, the BTZ black hole (and, equivalently, thermal $AdS_3$) can be understood as the quotient space $\mathcal{M}_\Gamma = \mathbb{H}^3/\Gamma$, with $\Gamma \simeq \mathbb{Z}$ being the group generated by (95). The Selberg zeta function $Z_\Gamma(s)$ for orbifold spaces of the form $\mathcal{M}_\Gamma = \mathbb{H}^n/\Gamma$ when $\Gamma$ is a Kleinian group was derived in [5], whereas [16] worked out the Selberg zeta function for the BTZ black hole associated with dilations of the Poincaré patch :

$$Z_\Gamma(s) = \prod_{k_1,k_2=0}^{\infty} \left[ 1 - e^{2ibk_1} e^{-2ibk_2} e^{-2a(k_1+k_2+s)} \right] , \tag{44}$$

Here $a = \pi\tau_2$, $b = -\pi\tau_1$, and $\tau = \tau_1 + i\tau_2$[8]. The infinite product converges for all $s$. $Z_\Gamma(s)$ is an entire function, with zeros occurring whenever the argument of the exponential equals $2\pi i\ell$ for $\ell \in \mathbb{Z}$. These zeros occur when $s = s^*_{\ell,k_1,k_2}$, with

$$s^*_{\ell,k_1,k_2} = -(k_1 + k_2) + \frac{ib}{a}(k_1 - k_2) - \frac{i\pi\ell}{a} . \tag{45}$$

Consider the quantity $\log Z_\Gamma(s)$. Expanding the logarithm in a Taylor series and performing two geometric series summations on $k_1$ and $k_2$, we arrive at

$$\log Z_\Gamma(s) = -\sum_{k=1}^{\infty} \frac{e^{-2\pi k\tau_2(s-1)}}{4k|\sin \pi k\tau|^2} . \tag{46}$$

Comparing (43) and (46), we obtain:

$$-\log \det \nabla_s^2 = \zeta'(0|\mathcal{O}) = \zeta'^{(e)} - 2\log Z_\Gamma(\Delta) , \tag{47}$$

where $\Delta = 1 + \sqrt{1 + m^2}$. This equation is our desired relationship between the Selberg and generalized Riemann zeta functions. We see that they are related through the divergent volume contribution $\zeta'^{(e)}$.

We can now elucidate the connection between the Selberg zeta function (and thus the heat kernel) and quasinormal modes. We repackage the integers $k_1$ and $k_2$ defined in (44) in terms of two new integers $j$ and $n$ as follows:

$$\begin{aligned} n \geq 0 : \quad & k_1 + k_2 = 2j + n \quad & k_1 - k_2 = \mp n , \\ n < 0 : \quad & k_1 + k_2 = 2j - n \quad & k_1 - k_2 = \mp n . \end{aligned} \tag{48}$$

These shifts produce four different combinations of $s^*$. For $n \geq 0$,

$$s^*_{\ell,n,j} = -(2j + n) - in\frac{b}{a} - i\frac{\pi\ell}{a} , \quad s^*_{\ell,n,j} = -(2j + n) + in\frac{b}{a} - i\frac{\pi\ell}{a} , \tag{49}$$

while for $n < 0$

$$s^*_{\ell,n,j} = -(2j - n) - in\frac{b}{a} - i\frac{\pi\ell}{a} , \quad s^*_{\ell,n,j} = -(2j - n) + in\frac{b}{a} - i\frac{\pi\ell}{a} . \tag{50}$$

The redefinitions (48) are not ad hoc. Remarkably, the prescription for defining $k_1$ and $k_2$ in terms of $j$ and $n$ is a result from scattering theory, and we urge the interested reader to see [16] for more details.

We will see below that $j$ and $n$ both have an interpretation in terms of quasinormal modes. In particular, $j$ becomes the radial quantum number, $n$ plays the role of the thermal $n$ in the Matsubara frequencies (with $n \geq 0$ corresponding to ingoing modes),

---

[8]For the BTZ black hole we have $a = \pi r_+$ and $b = \pi|r_-|$.

| Selberg zeta integers | Interpretation |
|---|---|
| $j$ (rewriting $k_1$ and $k_2$) | QNM radial quantum number $p$ |
| $n$ (rewriting $k_1$ and $k_2$) | Matsubara thermal integer $n$ |
| $\ell$ (condition that $Z_\Gamma(s^*) = 0$) | QNM angular quantum number $\ell$ |

Table 1: The dictionary associating the repackaging of Selberg zeta function integers and their interpretation in terms of quasinormal modes and Matsubara frequencies.

| Ingoing ($n \geq 0$) | Outgoing ($n < 0$) |
|---|---|
| $\omega_{\ell,j}^*(\Delta) = \ell - \frac{i}{\pi}(a + ib)(2j + \Delta)$ | $\omega_{\ell,j}^*(\Delta) = \ell + \frac{i}{\pi}(a + ib)(2j + \Delta)$ |
| $k_1 + k_2 = 2j + n \qquad k_1 - k_2 = n$ | $k_1 + k_2 = 2j - n \qquad k_1 - k_2 = -n$ |
| $\omega_{\ell,j}^*(\Delta) = -\ell - \frac{i}{\pi}(a - ib)(2j + \Delta)$ | $\omega_{\ell,j}^*(\Delta) = -\ell + \frac{i}{\pi}(a - ib)(2j + \Delta)$ |
| $k_1 + k_2 = 2j + n \qquad k_1 - k_2 = -n$ | $k_1 + k_2 = 2j - n \qquad k_1 - k_2 = n$ |

Table 2: Quasinormal modes and their associated Selberg zeta integer pairings.

and $\ell$ is the angular quantum number. For clarity and convenience, these relationships are recorded in Table 1.

We present evidence for the dictionary presented in Table 1 through three examples: a real scalar field, a massive spin-2 tensor field and a massive spin-1 vector field in a rotating BTZ black hole background [10]. The scalar case provides a simple example to illustrate the dictionary, and we will see that the graviton and vector examples are interesting in their own right.

The ingoing modes of a scalar field on a rotating BTZ background are (19)

$$\omega_{\ell,j}^*(\Delta) = -\ell - 2\pi i T_R(2j + \Delta) , \quad \omega_{\ell,j}^*(\Delta) = \ell - 2\pi i T_L(2j + \Delta) , \tag{51}$$

while the outgoing (antiquasinormal) modes are (20)

$$\omega_{\ell,j}^*(\Delta) = -\ell + 2\pi i T_R(2j + \Delta) , \quad \omega_{\ell,j}^*(\Delta) = \ell + 2\pi i T_L(2j + \Delta) , \tag{52}$$

where $j \in \mathbb{N}$ and $\ell \in \mathbb{Z}$. To obtain a result that is easily mapped to thermal AdS, we rewrite $\omega_{\ell,j}^*(\Delta)$ in terms of $a = \pi r_+$ and $b = \pi|r_-|$. Using $|r_-| = ir_-$, $T_R = \frac{a-ib}{2\pi^2}$ and $T_L = \frac{a+ib}{2\pi^2}$, the ingoing modes become

$$\omega_{\ell,j}^*(\Delta) = -\ell - \frac{i}{\pi}(a - ib)(2j + \Delta) , \quad \omega_{\ell,j}^*(\Delta) = \ell - \frac{i}{\pi}(a + ib)(2j + \Delta) , \tag{53}$$

while the outgoing modes become

$$\omega_{\ell,j}^*(\Delta) = -\ell + \frac{i}{\pi}(a - ib)(2j + \Delta) , \quad \omega_{\ell,j}^*(\Delta) = \ell + \frac{i}{\pi}(a + ib)(2j + \Delta) . \tag{54}$$

Each one of these expressions corresponds to a particular choice for the packaging of the Selberg zeta integers (48). The correct pairings are given in Table 2.

Now we validate the dictionary presented in Table 1. The quantity of interest will be the difference between the conformal dimension $\Delta$ of the field in question and the zeros of the Selberg zeta function $s^*_{\ell,k_1,k_2}$:

$$\Delta - s^*_{\ell,k_1,k_2} = \Delta + (k_1 + k_2) - \frac{ib}{a}(k_1 - k_2) + \frac{i\pi\ell}{a} \ . \tag{55}$$

As a concrete example, consider the ingoing mode

$$\omega^*_{\ell,j}(\Delta) = \ell - \frac{i}{\pi}(a + ib)(2j + \Delta) \ , \tag{56}$$

and its corresponding integer transformation

$$k_1 + k_2 = 2j + n \qquad k_1 - k_2 = n \ . \tag{57}$$

Combining (55)-(57), we arrive at:

$$\Delta - s^*_{n,\ell,j} = \frac{i\pi(\omega^*_{\ell,j}(\Delta) - \ell)}{(a + ib)} + \frac{(a - ib)}{a}n + \frac{i\pi\ell}{a} \ . \tag{58}$$

We learn that tuning $\Delta$ to the zeros of the Selberg zeta function is *equivalent* to tuning the quasinormal modes to the quantity:

$$\omega^*_{\ell,j}(\Delta) = \frac{in}{a\pi}(a^2 + b^2) - \frac{ib}{a}\ell \ , \tag{59}$$

The righthand side of this equation is exactly the Matsubara frequencies $\omega_n$ for a thermal field theory that includes rotation (as reported in (17)). No matter which quasinormal mode/integer transformation pair we choose from Table 2, we always arrive at Equation (59). Notice that when there is no rotation ($b = 0$), we recover the results reported in [20, 28]

$$\Delta - s^*_{n,\ell,j} = \frac{i\pi}{a}\omega^*_{\ell,j}(\Delta) + n \ . \tag{60}$$

Setting $\Delta = s^*_{n,\ell,j}$ leads to

$$\omega^*_{\ell,j}(\Delta) = -\frac{a}{i\pi}n = 2\pi iTn \ . \tag{61}$$

That is, the quasinormal modes $\omega^*_{\ell,j}(\Delta)$ are mapped to the Matsubara frequencies $\omega_n$ exactly when $\Delta$ is identified with the zeros of the Selberg zeta function. In summary, we see that the Selberg zeta function is the vehicle that takes us between the heat kernel and quasinormal mode methods of computing functional determinants.

## 3.2 A Formal Connection: Gravitons

We now write the 1-loop partition function for the massless graviton in terms of Selberg zeta functions, as in [27]; we extend this work by making the connection to quasinormal modes. The essential structure of the Selberg zeta function is independent of the field species, i.e., it only depends upon the background geometry. The primary difference comes from swapping the scalar conformal dimension $\Delta$ with, as we will show, "effective" conformal dimensions for the graviton and vector ghost.

First, we relate the 1-loop partition function for the graviton $Z^{(1)}_{\text{grav}}$ to the Selberg zeta function [16]

$$Z_\Gamma(\Delta) = \prod_{k_1,k_2=0}^{\infty} \left[1 - e^{2ibk_1}e^{-2ibk_2}e^{-2a(k_1+k_2+\Delta)}\right] = \prod_{k_1,k_2=0}^{\infty} \left[1 - q^{k_2/2}\bar{q}^{k_1/2}|q|^\Delta\right] \ , \tag{62}$$

just as we did for the scalar in (47). We start from the logarithm of the 1-loop partition function (36), and use $q^{2k} = e^{-2ka[2(1+ib/a)]}$ to find

$$
\begin{aligned}
\log Z_{\text{grav}}^{(1)} &= \sum_{k=1}^{\infty} \frac{e^{-2ka\Delta_2^{m_2<0}}}{k|1-q^k|^2} - \sum_{k=1}^{\infty} \frac{e^{-2ka\Delta_1^{m_1<0}}}{k|1-q^k|^2} + \text{c.c.} \\
&= \log \left( \frac{Z_\Gamma(\Delta_1^{m_1<0}) Z_\Gamma(\Delta_1^{m_1>0})}{Z_\Gamma(\Delta_2^{m_2<0}) Z_\Gamma(\Delta_2^{m_2>0})} \right) .
\end{aligned}
\tag{63}
$$

Here we have introduced

$$
\begin{aligned}
\Delta_2^{m_2<0} &= 2\left(1 + i\frac{b}{a}\right) = \Delta_2 - 2\frac{T_R - T_L}{T_R + T_L}, & \Delta_1^{m_1<0} &= 3 + i\frac{b}{a} = \Delta_1 - \frac{T_R - T_L}{T_R + T_L}, \\
\Delta_2^{m_2>0} &= 2\left(1 - i\frac{b}{a}\right) = \Delta_2 + 2\frac{T_R - T_L}{T_R + T_L}, & \Delta_1^{m_1>0} &= 3 - i\frac{b}{a} = \Delta_1 + \frac{T_R - T_L}{T_R + T_L}.
\end{aligned}
\tag{64}
$$

Following the notation in [10], here $\Delta_2 = |m_2| + 1$ and $\Delta_1 = |m_1| + 1$ are the conformal dimensions for a massive graviton and massive spin-1 ghost, respectively. The physical case of a massless graviton can be obtained by setting $m_2^2 = 1$ and $m_1^2 = 4$; we then find the physical values $\Delta_2 = 2$ and $\Delta_1 = 3$.

The quantities $\Delta_i^{m_i<0}, \Delta_i^{m_i>0}$ are "effective" conformal dimensions of a massive spin field. By effective, we mean that the condition $\omega_* = \omega_n$ can be brought to the same form as a real scalar field by shifting $\Delta_i \to \Delta_i^{m_i<0}$ or $\Delta_i^{m_i>0}$, as expressed in (64). This shift depends on the sign of $m_i$ as the quasinormal modes depend on the sign of $m_i$. We will show how the effective conformal dimensions arise from the condition $\omega_* = \omega_n$ momentarily,[9] but here we see the effective conformal dimensions naturally arise as the argument of the Selberg zeta functions.

As we observed for a real scalar field on the rotating BTZ black hole background in (59), when we tune the effective conformal dimensions $\Delta_2^{m_i<0}, \Delta_2^{m_i>0}$ to the zeros of the Selberg zeta function, we find that the quasinormal modes $\omega_*$ associated with the higher spin field in question are identified with the thermal (Matsubara) frequencies $\omega_n$.

First we show again how $\Delta_i^{m_i<0} = s^*$ leads to $\omega_* = \omega_n$ for a scalar, and then we use this language to prove the statement for the graviton and vector field. We begin by rewriting $s^*$ in terms of familiar quantities. For specificity, we work with the ingoing mode (49)

$$
s_{\ell,n,j}^* = -(2j+n) - in\frac{b}{a} - i\frac{\pi\ell}{a} .
\tag{65}
$$

and with positive angular momentum $k$. The other cases work very similarly. For positive $\ell$ we may express $s^*$ as

$$
s_{\ell,n,j}^* = -2j - n + ik_\Phi(n,\ell) ,
\tag{66}
$$

where we have used $k_\Phi(n,\ell)$ from (18). Then

$$
\Delta - s_{\ell,n,j}^* = 2j + \Delta + n - ik_\Phi(n,\ell) = 0 .
\tag{67}
$$

Identifying $j$ as the radial mode number, this equation is the condition that $\omega_n = \omega_*$, for the specific case of the ingoing quasinormal mode (56). The identical result is obtained upon choosing any mode in (49) and (50).

---

[9]This shift is also discussed in an appendix of [10], where it shown to arise from a regularity condition on the low-thermal number quasinormal modes for mode number $j$ less than the spin.

| Scalar | $2j + \Delta + n - ik_\Phi(n, \ell) = 0$ |
|---|---|
| Vector | $2j + \Delta_1 + (n+1) - ik_\Phi(n, \ell) = 0$ |
| Graviton | $2j + \Delta_2 + (n+2) - ik_\Phi(n, \ell) = 0$ |

Table 3: The condition $\omega_* = \omega_n$ for ingoing modes for scalar, vector and graviton fields with angular momentum quantum number $\ell$.

This relationship extends to the massive graviton and vector, as we now show. Concentrating again on the ingoing mode with postive angular momentum, and also specifying to the massive graviton with $m_2 < 0$ as in (64), we observe

$$\Delta_2^{m_2<0} - s_{\ell,n,j}^* = 2j + \Delta_2^{m_2<0} + n - ik_\Phi(n, \ell) = 0 \,. \tag{68}$$

We recover the condition $\omega_n = \omega_*$ for the graviton reported in Table 3 by shifting $n \to n+2$ in (68). The massive vector also works the same way. Specifying to $m_1 < 0$ and thus effective conformal dimension $\Delta_1^{m_1<0}$ as in (64), we find

$$\Delta_1^{m_1<0} - s_{\ell,n,j}^* = 2j + \Delta_1^{m_1<0} + n - ik_\Phi(n, \ell) = 0 \,. \tag{69}$$

This equation is also equivalent to $\omega_n = \omega_*$, as stated in Table 3, after shifting $n \to n+1$ in (69).

At first glance, it seems strange that the Selberg zeta function zeros provide access to the quasinormal modes of general spin $s$ fields, when the function itself depends only upon the spacetime, not the field content. However, while the Selberg zeros $s^*$ and the Matsubara frequencies $\omega_n$ are only dependent upon the spacetime, the effective conformal dimension $\Delta_i^{m_i}$ and the quasinormal mode frequencies $\omega_*$ depend on the spin of the perturbing field. Thus the statement that $\Delta_i^{m_i} - s^* = 0$ implies $\omega_* = \omega_n$ can hold for fields of any spin because the shift in $\Delta_i^{m_i}$ is accounted for by a corresponding shift in $\omega_*$.

We can generalize our expression of the 1-loop partition function in terms of Selberg zeta functions in (63) to other quotient geometries $\mathbb{H}^3/\Gamma$ for any discrete subgroup $\Gamma$ of the isometry group $PSL(2, \mathbb{C})$ of $\mathbb{H}^3$. The only change is an additional product over $\gamma \in \mathcal{P}$, where $\mathcal{P}$ is a set of representatives of the primitive conjugacy classes of $\gamma$:

$$Z_{(1),\text{grav}}^{\mathbb{H}^3/\Gamma} = \prod_{\gamma \in \mathcal{P}} \frac{Z_\Gamma^\gamma(\Delta_1^{m_1<0}) Z_\Gamma^\gamma(\Delta_1^{m_1>0})}{Z_\Gamma^\gamma(\Delta_2^{m_2<0}) Z_\Gamma^\gamma(\Delta_2^{m_2>0})} \,, \tag{70}$$

where

$$Z_\Gamma^\gamma(\Delta_\gamma) = \prod_{k_1,k_2=0}^{\infty} \left[ 1 - q_\gamma^{k_2/2} \bar{q}_\gamma^{k_1/2} |q_\gamma|^{\Delta_\gamma} \right] \,. \tag{71}$$

Here we made the substitution $q \to q_\gamma$ in (44), where $q_\gamma$ are the eigenvalues of primitive generators $\gamma$ of the discrete subgroup $\Gamma \subset SL(2, \mathbb{C})$ [7]. We emphasize that, in general, the effective conformal dimension $\Delta_\gamma$ will depend on the primitive $\gamma$.

## 4 Discussion

We have demonstrated the relationship between the heat kernel and quasinormal mode methods for calculating 1-loop determinants, both directly and formally. The examples we

considered were the scalar and graviton fields in a rotating BTZ black hole background. First, we showed that (1) the quasinormal mode sums build up the integrand of the heat kernel integral, and (2) the image sum of the heat kernel builds up the logarithms in the quasinormal mode expression. Building upon previous work, we demonstrate that the Selberg zeta function of a given quotient spacetime $\mathbb{H}^n/\Gamma$ acts as a bridge between the quasinormal mode and heat kernel methods. Tuning the effective conformal dimensions $\Delta_i^{m_i<0}$ or $\Delta_i^{m_i>0}$ to the zeros of the Selberg zeta function $s^*$ is equivalent to tuning the quasinormal modes $\omega_*$ to the Matsubara frequencies $\omega_n$ of the spacetime.

Using the formalism of Section 3, we may potentially predict quasinormal modes on more complicated quotient spacetimes $\mathbb{H}^n/\Gamma$, such as higher dimensional generalizations of the rotating BTZ black hole. As far as we know, quasinormal modes in such backgrounds are currently unavailable in the literature. If we know any two of (1) the Selberg zeta function of the spacetime, (2) the Matsubara frequencies and (3) the quasinormal modes, we can construct the third. We expect this predictive process to work for fields of all spin, including fermions.

Beyond considering higher dimensional manifolds of the form $\mathbb{H}^n/\Gamma$, we also hope to adapt our formalism to non-hyperbolic quotients $\mathcal{M}/\Gamma$ that possess sufficient symmetry. Product spacetimes may be a particularly interesting and straightforward extension, as many are expressible as quotients. We leave this for future work.

Since spacetimes of the form $\mathbb{H}^3/\Gamma$ are of interest in the study of holography, specifically when finding bulk quantum corrections to holographic entanglement entropy [29], one might hope that connections made in this work could circumvent challenges that arise in analytic calculations of 1-loop corrections to holographic Rényi entropies. However, it is difficult to express the eigenvalues of the Schottky generators $q$ in terms of boundary field data, so we leave this application to future work.

The relationship between quasinormal modes and the Selberg zeta function might further the understanding of quantum chaos in holographic field theories. Black hole quasinormal modes are dual to poles in the retarded two-point function of the corresponding operators in the dual field theory [23, 30]. Furthermore, if the field theory exhibits properties of chaos [31], the quasinormal modes are related via holography to the Ruelle resonances of the quantum theory [32], which control the decay of the two-point autocorrelation function of an operator $\mathcal{O}$. Indeed, Ruelle defined his own zeta function $\zeta_R(s)$, which generalized certain properties of the Selberg zeta function $\zeta_S(s)$ [33], and in the case of constant curvature manifolds is related to $\zeta_S(s)$ via [15, 34]:

$$\zeta_R(s) = \frac{\zeta_S(s)}{\zeta_S(s+1)} \ . \tag{72}$$

The zeros and poles of the Ruelle zeta function are related to the Pollicott–Ruelle resonances, while the singularities of the Selberg zeta function correspond to quasinormal modes and thus eigenvalues of the Laplacian (see [35] and references therein). The Selberg trace formula and zeta function also appear in the literature on chaos [15], and so perhaps the results presented here will have further applications in that subfield. We leave the study of both of these connections for future work.

While this work was in preparation, we became aware of another work which connects the standard heat kernel approach and normal/quasinormal mode method of computing 1-loop determinants [36]. While [36] connects these methods in the context of higher dimensional hyperbolic spacetimes using elements of Sturm-Liouville theory, we relate them in the context of $2+1$-dimensional hyperbolic quotient spacetimes via the Selberg zeta function. We expect our two complementary approaches will facilitate future work.

## Acknowledgements

We would like to thank Alejandra Castro, David McGady, Gim Seng Ng and Phillip Szepietowski for helpful discussions. The work of CK and VM is supported by the U.S. Department of Energy under grant number DE-SC0018330.

## A  BTZ Group Structure

In the body of this article we make extensive reference to the rotating BTZ black hole. Let us therefore briefly review the geometry of the BTZ black hole (for a longer review, see [37]). The rotating BTZ black hole in Boyer-Lindquist coordinates is given by

$$ds^2 = \frac{r^2}{(r^2 - r_+^2)(r^2 - r_-^2)}dr^2 - \frac{(r^2 - r_+^2)(r^2 - r_-^2)}{r^2}dt^2 + r^2\left(d\phi - \frac{r_+ r_-}{r^2}dt\right)^2 , \quad (73)$$

where we have set the AdS radius $L = 1$, and where the locations of the inner and outer horizons $r_-$ and $r_+$ are given by

$$r_+^2 + r_-^2 = M , \quad r_+^2 r_-^2 = \frac{J^2}{4} , \quad (74)$$

with $M$ and $J$ being the mass and angular momentum of the black hole, respectively. Note that we require the angular variable $\phi$ to be periodic $\phi \sim \phi + 2\pi$.

One useful form of the BTZ line element is

$$ds^2 = d\xi^2 - \sinh^2 \xi dT^2 + \cosh^2 \xi d\Phi^2 , \quad (75)$$

which can be arrived at by making the coordinate transformation

$$\tanh^2 \xi = \frac{r^2 - r_+^2}{r^2 - r_-^2} \quad T = r_+ t - r_- \phi \quad \Phi = r_+ \phi - r_- t . \quad (76)$$

We may also write the BTZ geometry in Poincaré patch coordinates

$$ds^2 = \frac{1}{z^2}(-dy^2 + dx^2 + dz^2) , \quad (77)$$

where

$$x = \tanh \xi \cosh T e^{\Phi} , \quad y = \tanh \xi \sinh T e^{\Phi} , \quad z = \operatorname{sech} \xi e^{\Phi} , \quad (78)$$

or equivalently,

$$x = \left(\frac{r^2 - r_+^2}{r^2 - r_-^2}\right)^{1/2} \cosh(r_+ t - r_- \phi)\exp(r_+ \phi - r_- t) ,$$

$$y = \left(\frac{r^2 - r_+^2}{r^2 - r_-^2}\right)^{1/2} \sinh(r_+ t - r_- \phi)\exp(r_+ \phi - r_- t) , \quad (79)$$

$$z = \left(\frac{r_+^2 - r_-^2}{r^2 - r_-^2}\right)^{1/2} \exp(r_+ \phi - r_- t) .$$

In this article we will be interested in the Euclidean BTZ black hole. Under the Euclideanization scheme $t \to -it_E$ and $J \to -iJ_E$, we have $r_- \to i|r_-|$, and, consequently,

$$T \to -iT_E , \quad \Phi \to \Phi , \quad (80)$$

and

$$x \to x , \quad y \to -iy_E , \quad z \to z . \tag{81}$$

In order for the coordinate transformation (79) to be regular at $r = r_+$ we require the identification

$$t_E \sim t_E + \beta_0 , \quad \phi \sim \phi + \theta , \tag{82}$$

where

$$\beta_0 = \frac{2\pi r_+}{r_+^2 - r_-^2} , \quad \theta = \frac{2\pi |r_-|}{r_+^2 - r_-^2} . \tag{83}$$

In the context of Euclidean quantum field theory, $\beta_0$ is interpreted as a finite temperature, while $\theta$ is the angular potential. This identification is equivalent to $T_E \sim T_E + 2\pi$, which allows the coordinate transformation (76) at $\xi = 0$ to be regular. It is often helpful to combine the two parameters $(\beta_0, \theta)$ into a single complex quantity $\tau = \frac{1}{2\pi}(\theta + i\beta_0)$.

The identifications

$$\phi \sim \phi + 2\pi , \quad (t_E, \phi) \sim (t_E + \beta_0, \phi + \theta) , \tag{84}$$

allow for the BTZ black hole to be understood as a quotient of $AdS_3$ by the set of integers $\mathbb{Z}$, where $\mathbb{Z}$ is generated by an element $\gamma \in SL(2, \mathbb{C})$, an isometry group of $\mathbb{H}^3$, such that

$$\gamma(y, w) \to (|q|^{-1}z, q^{-1}w) , \tag{85}$$

where we have defined the complex coordinate $w \equiv x + iy_E$ on the Euclideanized Poincaré patch of the BTZ black hole,

$$ds^2 = \frac{1}{z^2}(dz^2 + dw d\bar{w}) , \tag{86}$$

and introduced $q \equiv e^{2\pi i\tau}$. Geometrically we may picture the Euclidean space $\mathcal{M} = \mathbb{H}^3/\mathbb{Z}$ as a solid torus of constant negative curvature.

It is interesting see how each of the identifications (84) affect the coordinates $(x, y_E, z)$. A straightforward exercise reveals that $(t_E, \phi) \sim (t_E, \phi + 2\pi)$

$$\begin{aligned}
x &\to x' = [x\cos(2\pi|r_-|) - y_E \sin(2\pi|r_-|)] e^{2\pi r_+} \\
y_E &\to y_E' = [y_E \cos(2\pi|r_-|) + x\sin(2\pi|r_-|)] e^{2\pi r_+} , \\
z &\to z' = ze^{2\pi r_+}
\end{aligned} \tag{87}$$

while the $(t_E, \phi) \sim (t_E + \beta_0, \phi + \theta)$ behaves as the identity transformation:

$$x \sim x , \quad y_E \sim y_E , \quad z \sim z . \tag{88}$$

We can write these two transformations in a more suggestive form:

$$\begin{pmatrix} x' \\ y_E' \\ z' \end{pmatrix} = \gamma \begin{pmatrix} x \\ y_E \\ z \end{pmatrix} = \begin{pmatrix} e^{2a} & 0 & 0 \\ 0 & e^{2a} & 0 \\ 0 & 0 & e^{2a} \end{pmatrix} \begin{pmatrix} \cos 2b & -\sin 2b & 0 \\ \sin 2b & \cos 2b & 0 \\ 0 & 0 & 1 \end{pmatrix} \begin{pmatrix} x \\ y_E \\ z \end{pmatrix} , \tag{89}$$

with

$$2a \equiv r_+\theta - |r_-|\beta_0 = 0 , \quad 2b \equiv r_+\beta_0 + |r_-|\theta = 2\pi , \tag{90}$$

corresponding to the $(t_E, \phi) \sim (t_E + \beta_0, \phi + \theta)$ identification and

$$2a \equiv 2\pi r_+ \quad 2b \equiv 2\pi |r_-| , \tag{91}$$

associated with the $(t_E, \phi) \sim (t_E, \phi + 2\pi)$ identification. It is clear from here that $(t_E, \phi) \sim (t_E + \beta_0, \phi + \theta)$ identification can be understood as a identity transformation, while $(t_E, \phi) \sim (t_E, \phi + 2\pi)$ identification can be understood as a composition of a rotation in $\mathbb{R}^2$ with complex eigenvalues $e^{\pm 2ib}$, and a dilation $e^{2a}$. The above observation suggests a group-theoretic interpretation of coordinate transformations induced by the identifications of the BTZ black hole. Following [16], we can make this more precise.

Let $A \in SL(2, \mathbb{C})$. We say that $A$ is *loxodromic* when $\text{tr} A \in \mathbb{C}/\mathbb{R}$, i.e., if the entries of $A$ are real then $A$ is not loxodromic. Next, fix $a, b \in \mathbb{R}$ with $a \neq 0$ and define

$$\gamma = \gamma_{(a,b)} \equiv \begin{pmatrix} e^{a+ib} & 0 \\ 0 & e^{-(a+ib)} \end{pmatrix} \in SL(2, \mathbb{C}) . \tag{92}$$

By the above definition, $\gamma$ is loxodromic unless $b = \pi n$ for $n \in \mathbb{Z}$, in which case $\gamma$ is said to be *hyperbolic*. We can then define the discrete subgroup $\Gamma = \Gamma_{(a,b)} \in SL(2, \mathbb{C})$ as the group generated by $\gamma$ when $\gamma$ is loxodromic; precisely

$$\Gamma \equiv \{\gamma^n | n \in \mathbb{Z}\} . \tag{93}$$

In the context of the BTZ black hole, we see that the identification $(t_E, \phi) \sim (t_E, \phi + 2\pi)$ can be understood as the transformation generated by the elements of $\Gamma_{\text{BTZ}}$ with $a = \pi r_+$ and $b = \pi |r_-|$. Meanwhile, the identification $(\tau, \phi) \sim (\tau + \beta_0, \phi + \Phi)$, an identity transformation, has $a = 0$, and $b = 2\pi$, in which case the associated $\gamma$ is not loxodromic.

Next define the orbifold

$$\mathcal{M}_\Gamma = \mathbb{H}^3 / \Gamma . \tag{94}$$

The standard linear fractional action of $SL(2, \mathbb{C})$ on $\mathbb{H}^3$, under $\Gamma$, is restricted to

$$\gamma^n \cdot \begin{pmatrix} x \\ y \\ z \end{pmatrix} = \begin{pmatrix} e^{2an}(x \cos 2bn - y \sin 2bn) \\ e^{2an}(x \sin 2bn + y \cos 2bn) \\ e^{2an} z \end{pmatrix} . \tag{95}$$

This means that $\mathcal{M}_\Gamma$ is the space $\mathbb{H}^3$ with two points $p_1, p_2 \in \mathbb{H}^3$ identified if $p_1 = \gamma^n \cdot p_2$ for some $n \in \mathbb{Z}$.

The orbifold $\mathcal{M}_\Gamma$ may also be obtained from the fundamental domain $F$

$$F = \{(x, y, z) \in \mathbb{H}^3, \ 1 \leq \sqrt{x^2 + y^2 + z^2} \leq e^{2a}\} . \tag{96}$$

The fundamental domain is obtained by identifying points on the upper hemisphere of radius 1 with their images on the upper hemisphere of radius $e^{2a}$ under the action of the generator $\gamma_n$ of $\Gamma$ [17]. In this context, $a$ is understood as the length of the geodesic segment which projects to a single closed geodesic on the quotient $\mathcal{M}_\Gamma$.

Using $a > 0$, we find that the hyperbolic volume of the fundamental domain $F$,

$$\text{vol}(F) \equiv \int \frac{dx \, dy \, dz}{z^3} , \tag{97}$$

is infinite. One can show this explicitly using spherical coordinates – $x = \rho \cos \theta \cos \varphi, y = \rho \cos \theta \sin \varphi, z = \rho \sin \theta$ with $\theta \in [0, \pi/2]$ and $\varphi \in [0, 2\pi]$. When the volume of the fundamental domain of $F$ associated with orbifold $\mathcal{M}_\Gamma$ is infinite, $\Gamma$ is said to be a *Kleinian* subgroup of $SL(2, \mathbb{C})$.

Therefore, the Euclidean BTZ black hole with the given $\Gamma$ (93) can be understood as the orbifold $\mathcal{M}_{\Gamma_{\text{BTZ}}} = \mathbb{H}^3 / \Gamma_{\text{BTZ}} \simeq \mathbb{H}^3 / \mathbb{Z}$ for Kleinian group $\Gamma_{\text{BTZ}}$. In other words, the periodicity of $\phi$ means, in group theoretic terms, that the Euclidean BTZ black hole can be regarded as the quotient space $\mathcal{M}_{\Gamma_{\text{BTZ}}}$ under the action of $\Gamma_{\text{BTZ}}$ on $\mathbb{H}^3$.

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
