# Peer review of "Connecting quasinormal modes and heat kernels in 1-loop determinants"

_SciPost Physics_

## Round 1 · Referee Report · Anonymous (Referee 1) · 2019-10-23

Strengths

  1. The connection made in this paper of the quasi-normal modes of the BTZ black hole to the the Selberg zeta function.

Weaknesses

  1. Not exploiting this connection to a new situation.

Report

The authors discuss the construction of the one loop determinant
of fields in either the thermal or the BTZ background.
The construction is based on thinking of the one loop determinant
as a meromorphic function of the conformal dimension $\Delta$ corresponding
to the field. The poles of this function can be identified as the
quasi-normal modes of the filed.
This connection was first noted by reference [9].

The new point the authors are making is that
the one loop determinant can be identified with the
Selberg zeta function for the orbifold $H^3/\Gamma$ where
$\Gamma \sim Z$ for thermal AdS3 or the BTZ black hole.

This is achieved by re-writing the integers appearing in the
Selberg zeta function in terms of other integers which
occur in the quasi-normal modes (eq 48,for the scalars).
Summarized in Table1.

This is first done for the one loop determinant of the scalars and then for the gravitons which also involve vectors.

As such the Selberg zeta function does not contain the information
of the field. The authors point out the information of the field is
present by replacing the scalar conformal dimension to
an `effective' conformal dimension, which depends on the spin and mass
of the field.

The connection to the Selberg zeta function is interesting formally.
Practically it can lead to the determination of quasi-normal modes on
higher dimensional quotients as the authors mention or
the modes for other quotients of $H^3$

There are my comments.

In the case of higher spin fields, as the authors mention
the scalar conformal dimensions is replaced by `effective conformal dimension' which essentially shifts the ranges of integers.
Is this equivalent to the shifts seen in eq 5.8 of ref[22].
If so, the authors can mention this and then the generalisation
to arbitrary spin is quite straight forward. Here also the fact that the
dependence of the field in the one loop determinant arose just from these shifts.
The Selberg zeta function is also re-written with the
conformal dimension shifted appropriately.
I suggest the authors carry out the same analysis
for the arbitrary higher spin
case since it seems to me it is quite a straight forward extension.
The authors mention some words about it below paragraph below eq 69,
but they can be more explicit.

Consider quotients of H^3 is such that one obtains higher genus
boundaries or handle body geometries, quotients by Schottky groups.
The authors briefly mention
this connection around eq 70-71.
Now in such cases do the poles of the Selberg zeta function
correspond to `quasi-normal modes'?
Or is this identification only true when the discrete quotient
corresponds to either thermal AdS3 or the BTZ?
I suggest the authors comment on it, since these are the
generalisations where the connection pointed out by the authors
could indeed lead to new results.
There are analytical continuations of the quotients
by Schottky groups to
Minkowski signature discussed by Krasnov.

The paper would have been nicer if the authors indeed had identified
at least some of the poles of the Selberg zeta function
to modes in corresponding
to handle body geometries say for the scalar.

After the authors address these comments, the paper
can be considered for publishing.

Requested changes

The requested changes are written in the report.

  • validity: good
  • significance: ok
  • originality: ok
  • clarity: ok
  • formatting: good
  • grammar: good

Author:  Andrew Svesko  on 2019-11-15  [id 648]

(in reply to Report 1 on 2019-10-23)
Category:
answer to question
suggestion for further work

We thank the referee for their constructive and thought-provoking feedback. Below we will address their two editorial suggestions.

We agree with the author's comment that a higher spin generalization would be interesting. However, it turned out there were some interesting subtleties that arose and we saved this generalization for another work, which we recently posted (1910.07607). To briefly summarize, relabeling the Selberg integers $k_{1}$ and $k_{2}$ needs to be carefully generalized for higher spin fields. Specifically, we must ensure quasinormal modes Wick rotate to square-integrable Euclidean zero modes, a necessary condition in building the 1-loop partition function for higher spin fields.

We have added a paragraph below Eqn. (69) to provide some additional details about the subtleties of extending to higher spin, as well as a footnote in the conclusion to briefly mention the higher spin generalization.

The referee also suggested we comment on whether the ``poles of the Selberg zeta function correspond to `quasi-normal modes'" of spacetimes whose analytical continuations are quotients by Schottky groups, such as the black hole solutions discussed in Krasnov. Indeed, we believe the quasinormal mode method is sufficiently general that a quasinormal mode analysis of fields propagating on these g-handled black holes would allow us to build the 1-loop partition function of these more general hyperbolic quotients. To our knowledge, however, the Lorentzian quasinormal modes for such handle-bodied solutions are not known. In fact, while the Euclidean zero mode analysis on $\mathbb{H}^{3}/\Gamma$ is well known, it is ambiguous as to what the Euclidean zero modes Wick rotate to because it is unclear which of the cycles of the handle-body is to be identified as the thermal circle. It is natural to expect that when the arguments of the Selberg zeta function in Eq. (71) are tuned to its zeros, we may find a condition analogous to $\omega_{\text{Matsubara}}=\omega_{\text{quasi}}$. As such, our observations might be used to predict the quasinormal mode frequencies of the g-handled black holes discussed in Krasnov. To do this we would need to know the Matsubara frequencies and the Selberg zeta function of the hyperbolic quotient in question. It may be possible to make some headway when $\Gamma$ is a Schottky group. For example, in the case of computing the holographic entanglement entropy of two intervals on a line, the $q_{\gamma}$ in Eq. (71) are known explicitly to some order in an expansion in small cross ratio (as shown in Eq. (58) of (1306.4682). We have added a paragraph at the end of section 3 with these comments.

We hope that these clarifications will adequately satisfy the referee's request.

Attachment:

refereereply.pdf

---

## Round 1 · Referee Report · Anonymous (Referee 2) · 2019-10-28

Strengths

1-Heat kernel method and quasi-normal method are two different approaches to compute the one-loop determinant of a quantum field on a curved background. The paper addresses a natural connection between these two approaches.

2-The zeros of the Selberg zeta function in the complex plane determines the one-loop determinant as a function of the complex conformal dimension. Given Selberg zeta function and Matsubara frequencies, one can compute the quasi-normal frequencies.

3- This connection is particularly useful in the cases where quasi-normal frequencies are not known.

Weaknesses

1- It is not very clear whether this is a more efficient approach than the heat kernel or quasi-normal calculations. In particular, it would be interesting to see an example where heat kernel or quasi-normal calculations are not available but one can find the Selberg zeta function and predict quasi-normal modes.

Report

In this paper, the authors have looked at two apparently different approaches of computing the one-loop determinant of fields on the quotient of 3-dimensional hyperbolic spacetime. These are heat kernel method and quasi-normal method. The heat kernel method is very useful when the underlying space is a homogeneous space (or their quotient) and can be computed using the group-theoretic technique. The quasi-normal methods, on the other hand, require detailed knowledge of the complex frequencies for the background. It is known that the one-loop determinant can be computed using both the techniques.\\

The authors have found that the Selberg zeta function is a natural function that connects these two approaches. In particular, the zeros of the Selberg zeta function in the complex plane determines the one-loop determinant as a function of the complex conformal dimension. Furthermore, the quasi-normal modes at the zeros of the Selberg zeta function coincide with the Matsubara frequencies. The authors have then looked at the examples of scalar, vector, and graviton on the quotient of 3-dimensional hyperbolic spacetime.\\

The connection is very interesting and it deserved to be published. It is worth exploring further in higher dimensional spaces and for higher spin fields. In particular, for the higher spin fields on the hyperbolic spaces (and its quotient), there exist heat kernel computations. In these cases, it would be nice to find the Selberg zeta function and see such a connection.

Requested changes

None

---

## Editorial Decision

resubmitted